# The Role of Non-Canonical Hsp70s (Hsp110/Grp170) in Cancer

**DOI:** 10.3390/cells10020254

**Published:** 2021-01-28

**Authors:** Graham Chakafana, Addmore Shonhai

**Affiliations:** Department of Biochemistry, University of Venda, Private Bag X5050, 0950 Thohoyandou, South Africa

**Keywords:** Hsp110, Grp170, non-canonical Hsp70, chaperone, cancer

## Abstract

Although cancers account for over 16% of all global deaths annually, at present, no reliable therapies exist for most types of the disease. As protein folding facilitators, heat shock proteins (Hsps) play an important role in cancer development. Not surprisingly, Hsps are among leading anticancer drug targets. Generally, Hsp70s are divided into two main subtypes: canonical Hsp70 (*Escherichia coli* Hsp70/DnaK homologues) and the non-canonical (Hsp110 and Grp170) members. These two main Hsp70 groups are delineated from each other by distinct structural and functional specifications. Non-canonical Hsp70s are considered as holdase chaperones, while canonical Hsp70s are refoldases. This unique characteristic feature is mirrored by the distinct structural features of these two groups of chaperones. Hsp110/Grp170 members are larger as they possess an extended acidic insertion in their substrate binding domains. While the role of canonical Hsp70s in cancer has received a fair share of attention, the roles of non-canonical Hsp70s in cancer development has received less attention in comparison. In the current review, we discuss the structure-function features of non-canonical Hsp70s members and how these features impact their role in cancer development. We further mapped out their interactome and discussed the prospects of targeting these proteins in cancer therapy.

## 1. Introduction

Cancer accounts for approximately one sixth of total global annual deaths [1]. The most widely used interventions against cancer, such as chemotherapy and radiotherapy, are not always effective due to treatment-induced cellular, genetic and biochemical changes that often confer treatment resistance [2]. This, therefore, urgently necessitates the need to identify novel anticancer targets. Apart from their role as molecular chaperones, heat shock proteins (Hsps) play an important role in various cancer signaling pathways such as tumorigenesis, carcinogenesis and apoptosis [3,4]. As such, the role of Hsps as cancer biomarkers is increasingly becoming apparent.

Several proteins play crucial roles in the key hallmarks of tumorigenesis (Figure 1). It is therefore not surprising that upset of cellular proteostasis is one of the several factors that could drive tumor cell proliferation and metastasis. Tumor cells therefore rely on robust protein folding machinery to sustain conformational stabilities of the proteins required for their proliferation. As signaling molecules, Hsp members are intimately linked to cancer progression (Figure 1). For example, a small Hsp (sHsp), Hsp27 and Hsp70 inhibit release of cytochrome c, caspase 3 and 9 from the mitochondria, thus causing cells to evade apoptosis ([5,6], Figure 1). Several studies have indeed demonstrated the inactivation of the caspase cascade facilitated by Hsp27 binding with caspase 3 and cytochrome c released from mitochondria [7,8]. Furthermore, Hsp27 also induces resistance to chemotherapy by sequestrating cytochrome c upon its release from mitochondria into cytosol [8]. On the other hand, Hsp70 blocks apoptosis by binding to Bax to suppress its translocation to mitochondria [9], thus reducing permeabilization of the mitochondrial membrane, consequently, this inhibits release of mitochondrial apoptogenic molecules, such as cytochrome c [10].

Since they are constantly in a state of proteotoxic stress, cancer cells exploit Hsps to protect themselves against the toxic effects of aberrant oncoproteins, genomic instability, hypoxia and acidosis [11,12]. High Hsp expression levels are associated with poor prognosis and treatment resistance in cancer patients, since Hsps protect tumor cells from therapeutic stressors such as radiation and cytotoxic chemotherapy [13]. Indeed, overexpression of Hsps has been observed in a wide range of cancers, including breast, endometrial, ovarian, gastric, colon, lung and prostate cancer [13,14,15,16]. Due to their ability to oversee proteostasis, Hsps facilitate the folding and maturation of proteins involved in cancer signaling pathways (Figure 1). Therefore, elevated Hsp levels are often associated with tumor progression [4,17,18,19]. In addition to their proteostatic functions, some Hsps are translocated to the cell surface where they function as receptors, thus serving as conduits that regulate several cancer signaling pathways including cell proliferation and invasion [20]. The presence of Hsps on cancer cell surfaces is an aspect that warrants further research, as it presents opportunities for development of novel chemotherapeutic strategies. In addition, Hsps also act as chaperokines that trigger an immune response against oncogenic cells [21]. In a previous study, the Hsp70-derived peptide, TKD, was reported to stimulate natural killer (NK) cells which then targeted tumor cells harboring Hsp70 on their membranes [22,23]. This further represents a prospective Hsp-mediated anticancer intervention.

It has been demonstrated that inhibition of Hsp90 induces degradation of oncogenic proteins [24,25]. Not surprisingly, Hsp90 promotes the conformational maturation of several oncogenic signaling proteins, including steroid receptors such as p53 [26,27] and tyrosine kinases such as ErBb2 ([28], Figure 1). Additionally, the expression of some oncogenes in the absence of Hsp70 may result in cell inactivation [29]. This underscores the importance of Hsps in modulating oncogenic processes. Remarkably, Hsp70 and Hsp27 have both been shown to interact directly with protein intermediates of the apoptosis pathway [30,31,32]. Since it is highly expressed in malignant tumors and on the surface of tumor cells, Hsp70 typically serves as a biomarker of poor prognosis in cancer patients. Notably, the roles of the Hsp70 and Hsp90 in cancer development are becoming subjects of immense research interest [33,34]. Knockdown of the heat shock factor 1 (*HSF1*) gene coupled with Hsp90 inhibition was shown to disrupt cancer cell proliferation in vitro and tumor growth in vivo [35]. Furthermore, the same study demonstrated that *HSF1* knockdown combined with HSP90 inhibition not only facilitated the degradation of oncogenic proteins, but also induced cancer cell apoptosis and decreased activity of the ERK pathway [35]. Consequently, most Hsp-targeted anticancer treatment efforts have primarily focused on Hsp70 and its ER homologue, Grp78 as well as Hsp90 [36,37,38]. A kinase inhibitor, sorafenib, used in the treatment of renal cell carcinoma and hepatocellular carcinoma, is an example of an Hsp70 targeting anticancer drug which functions to reduce the expression of Grp78 in cancer cells [39]. Thus, small molecules that modulate Hsp expression as well as those that inhibit their activity constitute possible anticancer agents.

## 2. Hsp110/Grp170

The human genome encodes a total of 17 Hsp70s, four of which are Hsp110 or Grp170 protein homologues ([40]; Table 1). Grp170 proteins are closely related to the Hsp110 family of proteins which occur in the endoplasmic reticulum (ER) and are primarily induced by glucose deprivation [41]. Both Grp170 and Hsp110 proteins constitute a non-canonical clade of the Hsp70 family. Hence, in our narrative, except where it distinguishes the function of these two proteins within their distinct cellular localization, we use the terms Hsp110 and Grp170 interchangeably, as the two chaperones are generally similar in structure and function.

Hsp70s are typically characterized by an N-terminal nucleotide binding domain (NBD) and a C-terminal substrate binding domain (SBD), connected by a linker (Figure 2). Although the NBDs of canonical and non-canonical Hsp70s exhibit relatively high sequence conservation, their SBDs are more divergent [42]. In spite of their high conservation, members of the Hsp70 family are characterized by unique signature motifs that define their functional specialization within cells. Notably, Hsp110s are marked by extended acidic insertions located within their substrate binding domain, SDB-β and SBD-α subunits ([43], Figure 2A). Additionally, Hsp110s possess linker segments that are distinct from canonical Hsp70s [44,45]. Hsp110/Grp170 families also harbor unique TEDWYLEE motifs which distinguish them from canonical Hsp70s [41]. Although at present the actual functions of these motifs have not yet been determined, they are thought to facilitate co-chaperone or client protein binding in light of their location in the SBD. Interestingly, human Hsp110/Grp170 group members are delineated from each other by unique TEDWYLEE motifs [42].

Hsp110s possess seven β strands in the SBD, while canonical Hsp70s possess a total of eight β strands [43]. Using three-dimensional models, we also observed structural variations within the loop regions L_1,2_ and L_4,5_ of a human canonical Hsp70 (HspA1A) and Hsp110 (HspH1) (Figure 2). Structural variations arising within these SBD sections potentially account for the functional delineations of Hsp110s in comparison to canonical Hsp70s. Generally, the SBD of Hsp110s preferentially binds peptide substrates harboring aromatic residues in contrast to canonical Hsp70s, which preferentially bind substrates enriched with aliphatic residues of approximately seven residues in length [50]. Structural variations in loops of Hsp70 are important for the function of the chaperone. For example, loops L_1,2_ and L_3,4_ located in SBDβ and are thought to regulate substrate binding specificity [51]. It was recently reported that most variations in the SBD segments of Hsp70s not only occur within the loop regions of the substrate binding cleft but also in the helical lid (SBDα) sections [42]. Indeed, the SBDα segment of Hsp110 is endowed with acidic insertions that are absent in the canonical isoform (Figure 2). This suggests that the lid regulates functional specificity of Hsp70 [52]. Hsp110 is reported to possess significantly higher substrate binding efficiency than canonical Hsp70 [53] and this could be attributed to its longer SBDα lid segment. Furthermore, the yeast Hsp110 homologue, Sse1, was shown to exhibit unique peptide-binding preferences from the canonical Hsp70 homologue (Ssa1), suggesting that Hsp70 and Hsp110 substrates do not serve entirely overlapping functions [53]. The study demonstrated that Hsp110 does not have notable affinity for a typical Hsp70 peptide substrate (enriched with aliphatic residues), thus suggesting that Hsp70s and Hsp110s possess distinct peptide recognition motifs [53].

Using three-dimensional structural modeling tools (Biova Discovery Studio 4.5), we noted some structural differences between the four human Hsp110 isoforms (HspH1, HspH2, HspH3, Grp170). Generally, the NBD segments of the 4 non-canonical Hsp70s exhibit high conservation (Figure 2C). Conservation of the NBD of these non-canonical Hsp70s is important in light of the role of this motif in regulating nucleotide exchange of their canonical Hsp70 counterparts. However, notable variations exist within their substrate binding domains of the Hs110 types [51]. It is therefore conceivable that these variations observed within the individual loop segments may account for distinct substrate selectivity. As such, each of the Hsp110s may play distinct roles in chaperoning proteins involved in cancer signaling pathways. In comparison to Hsp110, Grp170 exhibits a unique alpha helical section within L_4,5_ and L_5,6_ (Figure 2C). Thus, as an endoplasmic reticulum-based chaperone, Grp170 is possibly functionally adapted for its role in the ER. Consequently, it is also conceivable that Grp170 chaperones a specialized set of oncogenic proteins located within the ER.

Functionally, Hsp110/Grp170 subfamily members bind misfolding polypeptides, to prevent their aggregation [44,54,55]. This way they maintain denatured protein substrates in a soluble, folding-competent state before handing them over to canonical Hsp70 for folding into the native state [56,57]. In addition, canonical Hsp70 releases its substrate in the presence of ATP and stably binds substrate in the ADP-bound state [58]. On the other hand, the chaperone function of Hsp110/Grp170 chaperones is not regulated by nucleotides, therefore, Hsp110 remains bound to substrate even in the ATP state [50]. Thus, Hsp110/Grp170 members are more effective holdase chaperones than their canonical Hsp70 counterparts [44,50,53]. Hence canonical Hsp70s serve as refoldases while Hsp110/Grp170 members are buffers against proteostatic stress [59,60]. Hsp110/Grp170 also function as nucleotide exchange factors (NEFs) of canonical Hsp70 [60,61].

Cancer cells are subjected to several forms of physiological insults, including anticancer interventions-induced stress, immune response, elevated reactive oxygen species, enhanced hypoxic and acidic conditions [62,63]. Under these conditions, the stress buffering role of Hsp110 becomes apparent. In vitro studies have indeed shown that Hsp110 is significantly more efficient at recognizing denatured proteins than canonical Hsp70 and is thus is more capable of salvaging misfolding-prone proteins during stress [43,50]. This feature may possibly be attributed to the fact that Hsp110 possesses a longer lid segment which enables it to more effectively bind substrates than canonical Hsp70. Similarly, the *Plasmodium falciparum* Hsp110 displayed better aggregation suppression activity than its canonical isoform in the presence of excess ATP [44]. Since the tumor milieu is generally in a critical state with respect to energy production [64], the ATP-independent nature of Hsp110 holdase chaperone function may be a crucial factor for cancer cell development. Indeed, Hsp110 was previously shown to be conformationally more stable to thermal stress than canonical Hsp70 [65]. This further confirms Hsp110 as a buffer against extreme cellular conditions that adversely impact proteostatic integrity. Notably, unlike its canonical counterpart, Hsp110 is largely allosterically dormant on account of its rigid linker, connecting the NBD to the SBD [66]. As such, Hsp110 would be expected to be conformationally more stable as a holdase chaperone in the wake of physiological changes [45].

## 3. Hsp110 Roles in Cancer Pathogenesis

Several factors that upset proteostasis, such as drug pressure, pH and temperature changes, threaten the survival of malignant cells. In response to physiological stress, cancer cells activate cytoprotective adaptive pathways in which Hsp110 expression is upregulated. Indeed, Hsp110 expression is reportedly upregulated in various cancers including melanoma, prolactinoma, pituitary adenoma, breast cancer, colorectal cancer, pancreatic cancer, thyroid cancer, esophageal cancer, lung cancer, bladder cancer, islet cell tumor, gastric cancer, lymphoma, seminoma and hepatocellular carcinomas [67,68,69,70,71,72]. Furthermore, high Hsp110 expression is a poor prognostic factor for patients with melanoma, esophageal cancer, gastric cancer, tongue squamous cell carcinoma, colorectal cancer, non-Hodgkin lymphoma, MDS or AML [67,73,74]. This may imply that suppression of Hsp110 expression in cancer cells may threaten the survival of tumor cells.

In cancer cells, Hsp110 may possibly facilitate protein stability and function by preventing aggregation of misfolded proteins as well as = maintaining protein conformation to enable ligand binding. A recent model proposed a possible role for Hsp110 in suppressing aggregation of α-synuclein [75]. The upregulation of α-synuclein is thought to contribute to aggressive phenotypes of meningiomas via the Akt/mTOR pathway, thus highlighting a possible role for Hsp110 in the development of malignant meningiomas by regulating the conformational stability of α-synuclein [76]. It is possible that α-synuclein may represent several Hsp110 clients implicated in malignant meningiomas, thus highlighting a crucial role of Hsp110 in shepherding conformational stability of the cancer cell proteome. Whether interception of Hsp110 function would abrogate progression of cancer remains to be established, however, such a prospect is worth investigating further.

A previous study demonstrated that HspH1 is a component of the β-catenin degradation complex which is implicated in cancer signaling [77]. HspH1 overexpression has also been shown to correlate with elevated nuclear β-catenin protein levels and upregulation of *Wnt* genes [77] implicated in cancer. As such, HspH1 upregulation accompanied with hyperactivation of Wnt signaling may pose as important prognostic biomarkers of cancer [78,79]. Notably, HspH1 overexpression is a prognostic biomarker that overall correlates with poor survival in breast cancer patients [77]. Whereas Hsp110 overexpression correlates with enhanced nuclear β-catenin protein levels and upregulation of *Wnt* gene, its depletion promoted hyperphosphorylation and degradation of β-catenin [77]. In light of the roles of Wnt and β-catenin in regulating transcription, the role of Hsp110 in overseeing the fate of these two molecules makes its role in cancer development apparent.

As a chaperone, Hsp110 confers thermotolerance to mammalian cells [43]. Furthermore, Hsp110 is involved in STAT3 phosphorylation in the cytosol, thereby promoting cell proliferation ([72], Figure 3). STAT3 is constitutively activated in several cancers and is thought to play a critical role in tumor growth and metastasis [80,81,82,83]. STAT3 also regulates several signaling pathways such as cellular proliferation, invasion and angiogenesis, which are all critical for metastasis [84,85]. As such, through its role in STAT3 phosphorylation, Hsp110 may regulate apoptosis and cancer development. Indeed, Hsp110 has been shown to protect cells from stress-induced apoptosis [71,86]. RNA interference targeting the *Hsp110* gene induced apoptosis in cancer cells, thus further pointing to an indirect role of this chaperone in the inhibition of apoptosis [87].

Aberrant cell migration is a major determinant of metastasis, leading to the development of malignant tumors [88]. As such, metastasis is the leading cause of cancer-related deaths [89]. Cancer cell migration and invasion into surrounding vasculature is a crucial initial step in metastasis [90]. Cell migration is a complex process characterized by several steps which include epithelial mesenchymal transition (EMT), abnormal angiogenesis and immune evasion ([91], Figure 3). During metastasis, cancer cells break free from the primary tumor to join the circulatory system, thus enabling colonization of distant organs. Interestingly, several protein molecules within the tumor microenvironment are associated with metastasis (Figure 3). Notably, Hsp110/Grp170 proteins play important roles in regulating activity of some proteins involved in these signaling pathways. Using a murine model, Manjili et al. [92] demonstrated that Hsp110 induces dendritic cells (DCs) to upregulate the expression of proinflammatory cytokines such as IL-6, TNF-*α* and IL-12. Similarly, tumor cells engineered to produce a secretable form of Grp170 triggered upregulated TNF-α secretion from DCs [93]. IL-6 and TNF-*α* play key roles in EMT, as previously demonstrated [92,94,95]. Upon EMT activation, tumor epithelial cells lose their cell polarity and adhesion properties to gain migratory and invasive properties, becoming mesenchymal cells [96,97]. Interestingly, the role of EMT in various cancers, including prostate, lung, liver, pancreatic and breast cancers, has been established [98,99]. Since Hsp110 is implicated in modulating proteins involved in EMT, its potential role in the development of metastasis could be inferred.

Angiogenesis is important in metastasis, as the growth and spread of neoplasms largely depends on the establishment of an adequate blood supply. Notably, Hsp110 potentially modulates angiogenesis. It has been established that Hsp110 co-operates with sHsps (such as HspB5) to suppress protein aggregation under stress conditions [100]. sHsp family members are known to modulate activity of the proangiogenic factor, VEGF, which induces structurally and functionally abnormal vasculature formation [101]. Hsp110 may therefore indirectly play a central role in angiogenesis and may thus constitute a promising target for novel anticancer therapy. T cells, monocytes and other immune cells are known to exert antimetastatic functions [102]. During the metastasic cascade, crosstalk between tumor cells and immune cells triggers immune evasion. This pathway is modulated by several anti-inflammatory cytokines such as transforming growth factor β (TGFβ), IL10 and IL35 [103,104]. Although the direct association of these cytokines with Hsp110 is yet to be experimentally validated, Hsp110 likely plays a key role in the folding of these proteins by canonical Hsp70s. Indeed, it has previously been reported that Hsp70s associate with and modulate function of some anti-inflammatory cytokines [105].

Hsp110 generally confers cytoprotection by functioning as a stress buffer which prevents stress-induced apoptosis. Previous studies have suggested that Hsp110′s antiapoptotic and chaperone roles are crucial for survival of tumor cells against the action of anticancer drugs or hypoxia [87]. Furthermore, Hsp110 upregulation suppresses cancer cell apoptosis by inhibiting the activation of caspase 9 and caspase 3 by blocking cytochrome c release from mitochondria [87,106,107]. Interestingly, the role of Hsp110 in activated B-cell diffuses large B-cell lymphoma (ABC-DLBCL) survival mechanisms has been also been established [108]. Hsp110 overexpression in ABC-DLBCL cell lines induces increased NF-kB signaling, thereby suggesting a tight interplay between Hsp110 and the NF-kB pathway [108]. This is particularly important since ABC-DLBCL tumors rely on sustained NF-kB activation for survival. At the intracellular level, Hsp110 possesses antiaggregation properties and also participates in the folding of nascent polypeptides or misfolded proteins in cells ([44], Figure 3). Further studies to elucidate the roles for Hsp110/Grp170 in cancer development are thus urgently required.

## 4. The Role of the ER Resident, Grp170 Chaperone in Cancer Pathophysiology

Since the ER is a critical organelle that facilitates several aspects of protein synthesis, including post-translational modification and subsequent folding, Grp170 plays a particularly significant role in cellular proteostasis. Like other Hsp110s, Grp170 generally exhibits dual functions; as an NEF for Grp78 (the ER Hsp70) and in aggregation suppression of secretory or transmembrane proteins in the ER [36]. The cytoprotective activity of intracellular Grp170 provides a survival benefit in cancer cells during tumor progression or metastasis [36]. Accumulating evidence demonstrates that Grp170 can directly bind to a variety of incompletely folded protein substrates in vivo in a nucleotide-independent fashion [109,110]. As such, Grp170 remains tightly bound to peptide substrates in both the ATP and ADP states, making it an efficient buffer against cellular stress [110].

Stress factors including glucose and oxygen deprivation within the tumor microenvironment are known to activate a Grp170-mediated unfolded protein response (UPR) to promote tumor cell survival [111]. Grp170 is thought to be a potential prognostic factor of breast cancer, since altered Grp170 levels correlate with different stages of tumor invasiveness [112,113]. Due to its ability to chaperone several proteins associated with cancer signaling pathways, Grp170 appears to possess pro-tumor activity ([114,115], Figure 3). In addition, Grp170 involvement in angiogenesis of tumors has been described through its ability to chaperone the major proangiogenic factor vascular endothelial growth factor (VEGF) [116,117]. Similarly, an antisense approach was used to demonstrate Grp170′s ability to reduce tumorigenicity in a prostate cancer model by blocking secretion of matured VEGF [117]. Grp170 has also been shown to associate with matrix metalloproteinase-2 (MMP-2) thereby promoting tumor invasion [118].

In as much as additional studies are necessary to glean a better understanding of the precise mechanistic contribution of the non-canonical Hsp70s in tumorigenesis, their chaperoning function appears to account for their pro-tumor activity. Complete proteomic studies on the involvement of Hsp110/Grp170 in cancer pathophysiology are worth exploring. Notably, Hsp110/Grp170 are predicted to interact with a large complement of proteins that are implicated in cancer development (Figure 3).

## 5. Identification of Potential Unique Proteomic Signatures of Hsp110/Grp170

We predicted the interactome of Hsp110/Grp170 homologues using the STRING 10.5 database ([119], http://string-db.org/). The predicted interactomes of the proteins revealed possible associations of these chaperones with several proteins implicated in tumorigenesis (Table 2). Generally, while there were overlapping interaction partners between the various Hsp110 forms, we noted that the chaperones were also marked with possibly unique interactomes (Figure 4). For instance, HspH2 and Grp170 were predicted to interact with a large complement of protein modifying enzymes as opposed to HspH1 and HsapH3 (Figure 4). Additionally, Grp170 also seemed to interact with a large complement of proteins that are involved in several other roles including protein translocation. The observed variations in interactomes may possibly arise from the structural variations and ER localization of Grp170, which makes it functionally specialized for binding ER proteins. Seemingly, the different Hsp110 isoforms play unique roles in chaperoning proteins involved in the different cancer signaling pathways as described below.

Notably, all the Hsp110 isoforms (HspH1, HspH2, HspH3 and Grp170) were predicted to associated with the cyclin G dependent kinase, GAK (Table 2). Cyclin dependent kinases are key regulatory enzymes that are involved in cell proliferation, which is an important hallmark of tumorigenesis. Previously, it has been established that GAK enhances the androgen receptor (AR) transcriptional response in androgen-independent prostate cancer [120]. Furthermore, GAK has been proposed as a druggable anticancer candidate that has broad therapeutic applications across numerous tumor types, including breast and colorectal cancers [121]. Given its important role in maintaining GAK in a functional state, it is therefore conceivable that Hsp110s have a crucial role in promoting tumorigenesis.

Intriguingly, Hsp110 isoforms, HspH3 and HspH2, were predicted to interact with a large complement of nucleoporins as opposed to Grp170 and HspH1. Nucleoporins are components of nuclear pore complexes (NPCs) which are huge macromolecular assemblies in the nuclear envelope, through which bidirectional cargo movement between the nucleus and cytoplasm occurs [122]. Several nucleoporins are linked to cancer, mostly in the context of chromosomal translocations, which encode nucleoporin chimeras [123]. Tumor cells are thought to exploit specific properties of nucleoporins to deregulate transcription, chromatin boundaries and essential transport-dependent regulatory circuits [123]. The nucleoporin POM121, which is predicted to interact with HspH3 and Hsp2, has reportedly been linked to prostate cancer [124]. POM121 has also been reported as a novel prognostic marker of oral squamous cell carcinoma [125]. It is therefore plausible that HspH3 and HspH2 play crucial roles in chaperoning POM121 in cancer progression. Several other nucleoporins, including translocated promoter region (TPR), Nup98 and Nup214, were also predicted to interact with HspH3 and HspH2 (Table 2). These proteins have previously been described as “promiscuous nucleoporins” due to their unique ability to associate with various partners to produce a variety of oncogenic fusion proteins [126]. Thus, HspH3- and HspH2-directed therapies may also hold prospects in prostate cancer intervention.

It was also predicted that HspH3 and Grp170 interact with EDEM3, whose upregulation is linked to thyroid cancer (Table 2). It remains to be established if the enhanced expression of EDEM3 associated with thyroid cancer is accompanied with a concomitant increase in HspH3 levels. The possibility of HspH3 as a biomarker for thyroid cancer is therefore worth exploring. Previous evidence suggests that, SUMOylation is implicated in cancer cell signaling and gene networks that regulate carcinogenesis, proliferation, metastasis and apoptosis [127]. Interestingly, HspH3 was predicted to interact with the SUMO protein, RANBP2 (Figure 4, Table 2). This implies that the chaperone potentially modulates SUMOylation in cells, possibly resulting in tumorigenesis.

Sec proteins form part of the heterotrimeric Sec61 and the dimeric Sec62/Sec63 complexes located in the ER membrane [128]. These complexes are thought to play a central role in the translocation of nascent and newly synthesized precursor polypeptides into the ER. Notably, Sec overexpression has been linked to cancer. Interestingly, several Sec proteins were predicted to interact with Hsp110 and Grp170 chaperones (Table 2). In a study conducted by Diwadkar et al. [129], interbreeding of *Sec tRNA* transgenic mice with a model of prostate cancer resulted in accelerated development of prostatic intraepithelial neoplasia (PIN) and more aggressive high-grade lesions.

## 6. Heroes or Villains: The Role of Heat Shock Proteins in Preventing Cancer Progression

Primarily, as chaperones, the role of Hsps is to assess the folding status of a protein towards assisting its refolding or, alternatively, to channel it towards degradation. In this way, Hsps may either hide metabolic consequences of mutations or they may expose them. More intriguingly, Hsps are thought to serve as buffers of protein mutations, i.e., they enable otherwise mutated proteins to fold into their functional conformations [130]. Thus, depending on the pathway that their respective substrates are implicated in, Hsps may promote or obstruct cancer development. However, on the balance it appears that malfunctioning of chaperones leads to general deregulation of several metabolic pathways, including those involved in signal transduction [130].

Mechanistically, Hsp110 functions as part of a protein assembly (chaperome) in concert with several other chaperones such as Hsp70 and sHsps to facilitate cellular proteostasis (Figure 5). Hsp110 chaperome complexes constitute multimolecular scaffolds that exhibit functional versatility to cater to the sustained function of the oncoprotein complement of cancer cells [131]. Interestingly, cell stress is thought to further stabilize chaperome complexes by reducing the dynamic nature of chaperone–chaperone/co-chaperone interactions ([131]; Figure 5). Alterations in protein–protein interaction networks are thought to be at the core of malignant transformation [132]. It is also thought that stress may remodel the proteome by both enhancing of interaction affinity of the chaperone complement as well as by switching interaction partner combinations, thus leading to proteome-wide connectivity dysfunctions [133]. How stress modulates a chaperome to alter protein–protein interactions in cancer remains yet to be fully explored. It was recently established that N-glycosylation of the ER-resident Hsp90 chaperone, Grp94, during stress led to global remodelling of cellular protein networks [133]. 

Hsp110 is known to directly associate with canonical Hsp70 through NBD–NBD interactions ([44,54,61], Figure 5) to form a heterodimer that facilitates Hsp70-based protein folding. Hsp110 acts a co-chaperone of Hsp70, facilitating the exchange of ADP for ATP state, leading to substrate release from the canonical Hsp70 partner (Figure 5). Given that Hsp70 facilitates the folding of virtually all proteins within the cell [134], the NEF role of Hsp110 in this regard is crucial to the efficient function of canonical Hsp70.

Hsp110s likely play multiple roles in cancer cells, since their chaperone function involves the stabilization of oncogenic proteins and those involved in cancer signaling pathways. The emerging role of Hsp110 in immunomodulation has been described [72,92] and in this regard, Hsp110 is thought to trigger inflammatory responses that propagate tumor growth. Thus, Hsp110 may play a role in tumorigenesis through its chaperokine role. It was recently demonstrated that tumors associated with ER-stress relay stress signals to neighboring cells through secretion of soluble mediators, triggering an exaggerated inflammatory response that facilitates tumor progression [135]. Although it is not clear as to whether Grp170 directly contributes to this inflammatory response induced pro-tumor effect, its involvement in this pathway is highly probable.

Previous studies have shown that Hsp110s and Grp170 are immunogenic chaperones [36,136]. As such, these chaperones are promising cancer vaccine candidates. In a murine cervical cancer model, Hsp110 was shown to not only improve the antitumor efficacy of cytotoxic T-lymphocyte epitope E7, but it also significantly inhibited tumor growth [137]. In addition, Hsp110 and carbonic anhydrase IX, of which the latter is the renal cell carcinoma-specific tumor protein, were shown to inhibit growth of renal cell carcinoma in mice [138]. This may imply that Hsp110 plays important roles in activating carbonic anhydrase IX. Hsp110-HER2 complex based vaccines were also shown to induce immune protection against spontaneous breast tumors in a transgenic mice model [139]. Grp170-based anticancer immunotherapeutic approaches are beginning to be explored. It has been demonstrated that a complex of Grp170 and tumor protein antigens activated the immune response, leading to inhibition of tumor growth in a melanoma mouse model [140]. Furthermore, mouse prostate cancer cells engineered to effectively secrete Grp170 exhibited enhanced tumor immunogenicity and cytolytic activity of distant tumors [141]. These studies provide evidence of the immunomodulatory roles of Grp170, possibly presenting them as potential vaccine candidates.

Hsp110 was shown to inhibit activation of dendritic cells through scavenger receptor binding [114]. Notably, Hsp110 has been described as the main chaperone involved in colorectal tumorigenesis and the presence of an Hsp110 inactivating mutation is directly associated with a good prognosis [142]. Interestingly, increased Hsp110 expression has been linked to tumor immunosurveillance [72]. A study by Berthenet et al. [72] demonstrated that Hsp110 overexpression in colorectal cancer cells induces the formation of macrophages with an anti-inflammatory profile. Although the precise mechanisms underlying extracellular release of Hsps (active vs. passive) remain speculative, increased Hsp levels are generally observed in the tumor microenvironment [143]. Indeed, several studies have demonstrated the anti-inflammatory role of extracellular Hsps and, depending on their levels, Hsps could either promote inflammation or suppress it [144]. In this regard, Hsp110/Grp170 may thus play a signal transducer role. Although several studies have demonstrated that Hsp110/Grp170 overexpression triggers release of proinflammatory cytokines, the effect of Hsp110 downregulation on the overall cancer cell proteome remains to be explored. A recent study showed that Hsp110 downregulation is detrimental to colon cancer cell proliferation [145]. This study suggests that Hsp110 depletion or inhibition may have detrimental effects on the proteome of the cancer cell.

## 7. Could Hsp110/Grp170 Be Targeted in Cancer Therapy?

Owing to their prominent role as chaperones, Hsp110 and Grp170 could serve as novel chemotherapeutic targets against cancer. Small molecule inhibitors such as polymyxin B (PMB) and epigallocatechin gallate (EGCG) possess great potential in this regard, as they have been successfully used to inhibit the activity of the *Plasmodium falciparum* Hsp110 in vitro [146]. These two compounds bind to the NBD, thus abrogating the basal ATPase activity of the Hsp110. However, since the holdase chaperone function of Hsp110/Grp170 is not modulated by nucleotides, targeting the Hsp110 NBD using nucleotide mimetics might not interfere with their direct chaperone role. However, the same NBD-targeted drugs may disrupt the NEF function of these chaperones which in turn would adversely impact fold and function of several proteins implicated in cancer development. Selectively targeting the SBD of Hsp110/Grp170 using peptide substrate mimetics may be an alternative approach. Two drug candidates targeting the SBD of Hsp70, 2-phenylethynesulfonamide (PES) and the TKD-motif directed peptide inhibitor, cmHsp70.1 have entered the clinical trials stage [147,148,149]. The design of domain-specific inhibitory compounds may prove useful in Hsp110-targeted anticancer therapy (Figure 6). Since Hsp110/Grp170 functions in co-operation with several other Hsps, its inhibition may impact the folding fate of the protein complement that drives cancer development and progression. In a recent study, Gozzi [145] designed a novel NBD-binding small molecule inhibitor which compromises Hsp110 chaperone function, thereby inhibiting STAT3 phosphorylation and colorectal cancer cell proliferation. There is therefore an urgent need for identification of novel compounds that target Hsp110/Grp170 chaperones in the fight against cancer. Apart from chemical compounds, antibodies and aptamers could alternatively be designed towards abrogating Hsp110/Grp170 functions. In a colon cancer murine model, an Hsp70 monoclonal antibody-based inhibitor, cmHsp70.1, which binds the TKD motif, was shown to significantly reduce tumor weight and improve survival rate [150]. One of the main challenges in the design of anticancer agents is to develop compounds that are safe. The varied proteomic composition of cancer cells compared to normal cells makes the inhibition of Hsps promising in light of their role as custodians of proteostasis.

## 8. Conclusions

Recently, Hsp110/Grp170 chaperones have emerged to a focal point in prospecting for novel chemotherapeutic targets, mainly due to their central roles in both proteostasis and signaling pathways. As nucleotide-independent holdase chaperones, Hsp110/Grp170 are regarded as cellular buffers against proteostatic stress. It is thus not surprising that their role in the cytoprotection of tumor cells, particularly in response to both drug- and hypoxic-stress, is becoming apparent. This review explored the possible interactome of these proteins and established that molecules involved in cancer development are amongst some of their most distinct clientele. This, coupled with their correlated expression with cancer prognosis, suggests a crucial role for these chaperones in cancer development. It is thus envisaged that targeting these group of chaperones has potential as an intervention tool against cancer.

## Figures and Tables

**Figure 1 cells-10-00254-f001:**
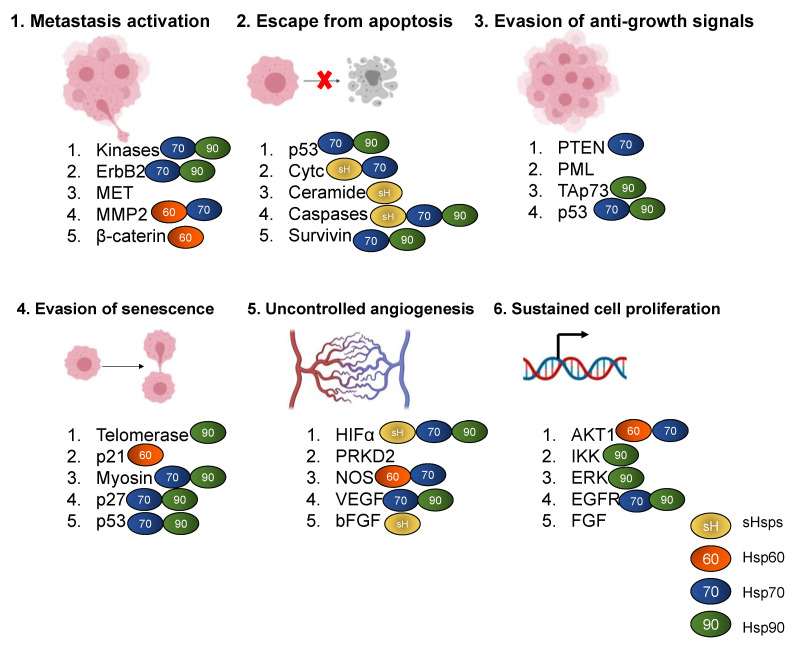
The proteomic landscape of the hallmarks of cancer: The processes of metastasis, uncontrolled angiogenesis, evasion of anti-growth signals, escape from apoptosis, cell proliferation and evasion of senescence are all crucial to tumor cell growth. Each of these processes is regulated by several proteins that play important roles in the respective signaling pathways. As chaperones, sHsps, Hsp70, Hsp90 and Hsp60 associate with several proteins that regulate cancer signaling pathways.

**Figure 2 cells-10-00254-f002:**
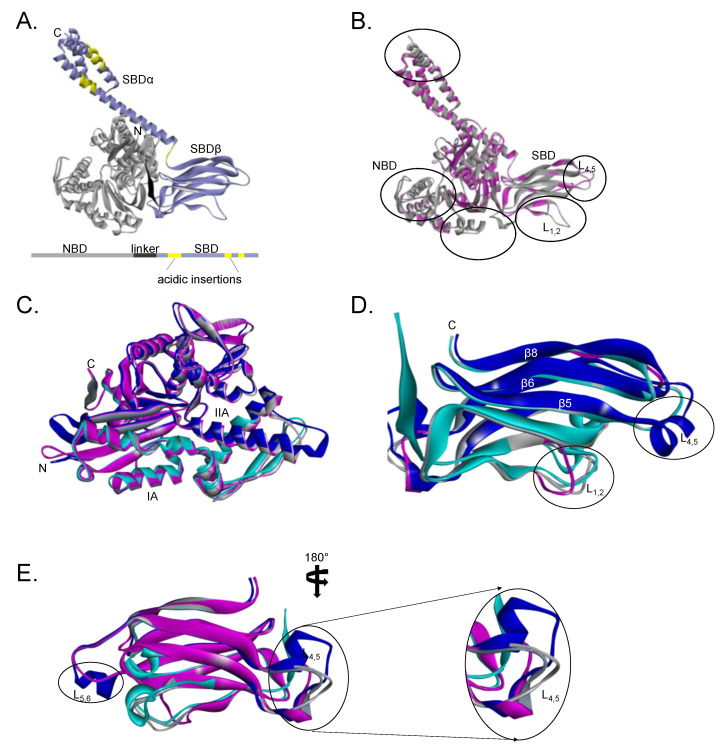
General structural features of human Hsp110s: (**A**) Structure of Hsp110 showing major features, including the unique acidic insertions in the substrate binding domain (SBD) α region. (**B**) Comparative structural analyses of a canonical Hsp70 (HspA1, purple) and an Hsp110 (HspH1, grey). Variations are predicted to occur within L_1,2_ and L_4,5_. (**C**) The nucleotide binding domains (NBDs) of human HspH1 (cyan), HspH2 (purple), HspH3 (grey) and Grp170 (blue) show high conservation. (**D**) The SBDs of human HspH1 (cyan), HspH2 (purple), HspH3 (grey) and Grp170 (blue) show variation within the SBDβ segments. (**E**) Major variations are predicted to occur at loops L4,5 and L5,6 of the SBDβ segments of human HspH1 (cyan), HspH2 (purple), HspH3 (grey) and Grp170 (blue).

**Figure 3 cells-10-00254-f003:**
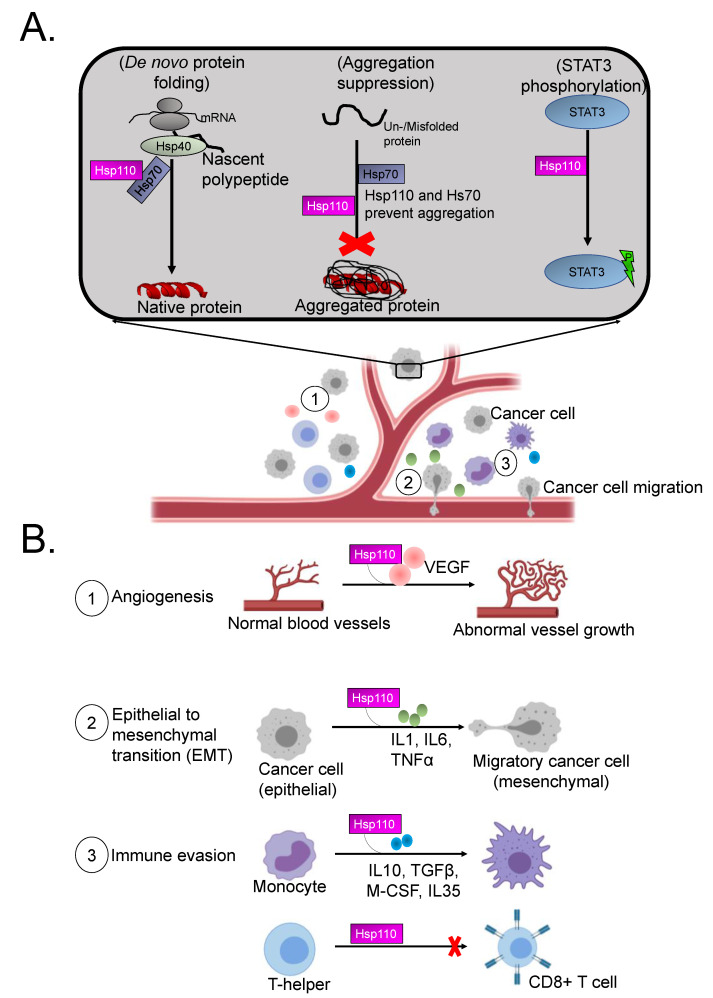
Roles of Hsp110 in cancer cells: (**A**) Hsp110 co-operates with Hsp70 complexed to its co-chaperone Hsp40 to facilitate folding of nascent polypeptides in tumor cells. Hsp110 also independently suppresses aggregation of oncogenic proteins and regulates functional competence of molecules such as STAT3 whose phosphorylation it facilitates, thereby promoting tumor cell proliferation. (**B**) Hsp110 modulates metastasis in tumour cells via interaction with VEGF leading to abnormal angiogenesis (1). Hsp110 is also involved in EMT via association with proinflammatory cytokines (2) and in immune evasion pathways (3) which ultimately result in metastasis.

**Figure 4 cells-10-00254-f004:**
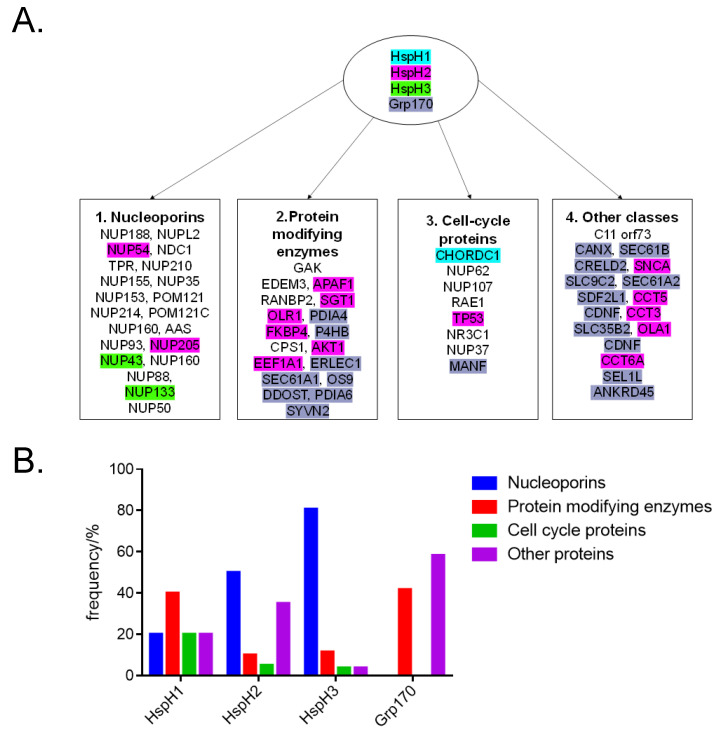
Predicted clients of Hsp110 and Grp170 members: (**A**) The predicted clientele of HSP110 and Grp170 are shown. Client proteins that are unique to each Hsp110/Grp170 type are shown highlighted in blue (HspH1), purple (HspH2), green (HspH3) and green (Grp170). Client proteins that are shared by more than one Hsp110/Grp170 type are not highlighted. Predictions were conducted using STRING database at a cut off score of 0.75. (**B**) Bar graph showing relative frequencies of clients of Hsp110 and Grp170 chaperones. While HspH1, HspH2 and HspH3 are generally predicted to associate with nucleoporins and cell-cycle regulating proteins, Grp170 interacts with a large complement of protein modifying enzymes and proteins involved in other functions such as translocation.

**Figure 5 cells-10-00254-f005:**
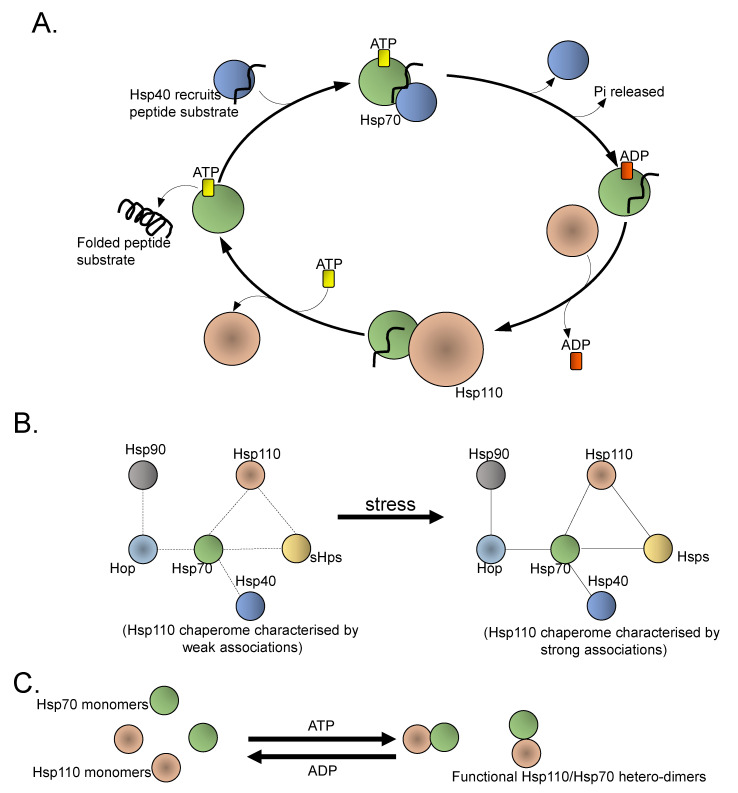
Hsp110 as an essential oncogenic proteome regulating chaperome: (**A**) Hsp110s facilitate exchange of ADP to ATP by canonical Hsp70, allowing release of a folded peptide. (**B**) Affinity of partners of the Hsp110 chaperome is enhanced by stress. (**C**) The formation of functional Hsp110:Hsp70 heterodimers is modulated by nucleotides.

**Figure 6 cells-10-00254-f006:**
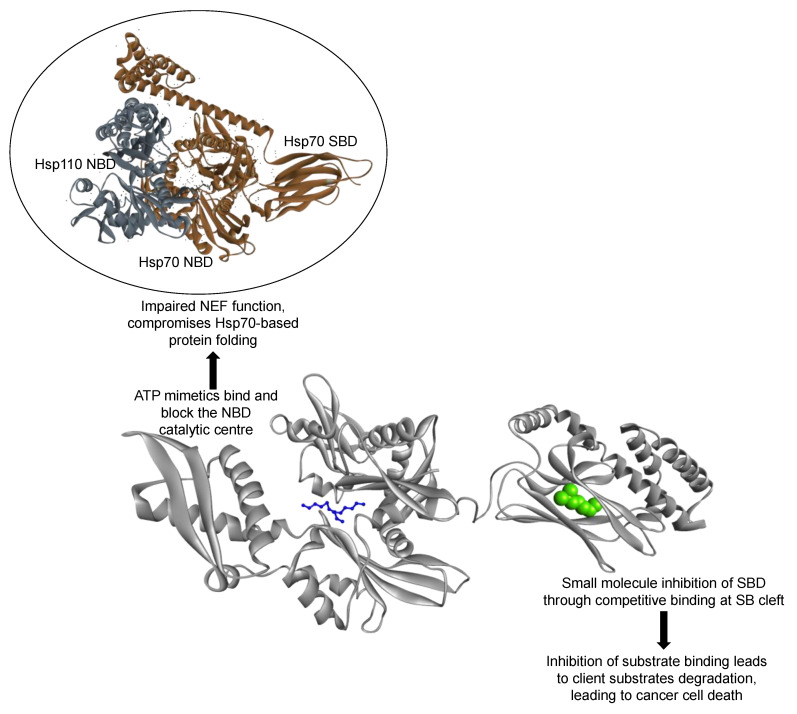
Proposed strategies for targeting Hsp110: Small molecule inhibitors that bind and block the NBD of Hsp110 abrogate its nucleotide exchange function on canonical Hsp70. Peptide mimetics also constitute possible Hsp110 inhibitors that abrogate its chaperone function.

**Table 1 cells-10-00254-t001:** Hsp110/Grp170 proteins of human origin.

Protein (Accession Number)	Size (kDa)	Localization	Stress Inducible (Yes/No)	Cellular Functions	References
1. HspH1 (Q92598)	97	Cytosol, nucleus, endocytic vesicle	Yes	Apoptosis suppression, aggregation suppression, nucleotide exchange factor (NEF)	[46]
2. HspH3 (O95757)	95	Cytosol, nucleus	Yes	Ellicits humoral immune responses in leukemia patients	[47]
3. HspH2 (P34932)	95	Cytosol, extracellular exosome	ND	Implicated in spermatogenesis	[48]
4. Grp170 (Q9Y4L1)	111	ER	Yes	Aggregation suppression, NEF	[49]

ND: not determined.

**Table 2 cells-10-00254-t002:** Predicted interaction of Hsp110/Grp170 with cancer-associated proteins.

Protein	Function	Score
HspH1 Interaction Partners		
1. GAK-cyclin G (kinase)	Associates with cyclin G and CDK5 and is involved in the uncoating of clathrin-coated vesicles by Hsc70.	0.879
2. CPS1(Carbamoyl-phosphate synthase)	Involved in the urea cycle and plays an important role in removing excess ammonia from the cell.	0.874
3. EDEM3 (ER degradation-enhancing alpha-mannosidase-like protein 3	Accelerates ER-associated degradation (ERAD) of glycoproteins by proteasomes.	0.757
4. CHORDC1 (Cysteine and histidine-rich domain-containing protein 1)	Regulates centrosome duplication, probably by inhibiting the kinase activity of ROCK2. Proposed to act as co-chaperone for HSP90. Prevents tumorigenesis.	0.747
HspH3 Interaction Partners		
1. NUP188 (Nucleoporin)	May function as a component of the nuclear pore complex (NPC).	0.937
2. C11 orf73	Acts as a specific nuclear import carrier for HSP70.	0.931
3. NUP37 (Nucleoporin)	Component of the Nup107-160 subcomplex of the nuclear pore complex (NPC) required for normal kinetochore microtubule attachment, mitotic progression and chromosome segregation.	0.926
4. RANBP2 (E3 SUMO-protein ligase)	Facilitates SUMO1 and SUMO2 conjugation, (Ran-GTP, karyopherin)-mediated protein import. Component of the nuclear export pathway.	0.924
5. TPR (Nucleoprotein TPR)	Essential for normal nucleocytoplasmic transport of proteins and mRNAs, plays a role in the establishment of nuclear-peripheral chromatin compartmentalization in interphase, and in the mitotic spindle checkpoint signaling during mitosis.	0.917
7. RAE1 (mRNA export factor)	Plays a role in mitotic bipolar spindle formation. May function in nucleocytoplasmic transport.	0.908
8. NUP155 (Nuclear pore complex protein)	May be essential for embryogenesis. Nucleoporins may be involved both in binding and translocating proteins during nucleocytoplasmic transport.	0.907
9. NUP153 (Nuclear pore complex protein)	Essential for normal nucleocytoplasmic transport of proteins and mRNAs. Involved in the quality control and retention of unspliced mRNAs in the nucleus.	0.904
10. NUP214 (Nuclear pore complex protein)	May serve as a docking site in the receptor-mediated import of substrates across the nuclear pore complex.	0.904
11. NUP62 (Nuclear pore glycoprotein)	Plays a role in mitotic cell cycle progression by regulating centrosome segregation, centriole maturation and spindle orientation. It might be involved in protein recruitment to the centrosome after nuclear breakdown.	0.904
12. NUP93 (Nuclear pore complex protein)	During renal development, regulates podocyte migration and proliferation through SMAD4 signaling.	0.904
13. NUP43 (Nucleoporin)	Component of the Nup107-160 subcomplex of the nuclear pore complex (NPC) required for normal kinetochore microtubule attachment, mitotic progression and chromosome segregation.	0.903
14. NUP88 (Nuclear pore complex protein)	Essential component of nuclear pore complex.	0.903
15. NUP133 (Nuclear pore complex protein)	Involved in poly(A)+ RNA transport.	0.903
16. NUP50 (Nuclear pore complex protein)	Interacts with regulatory proteins of cell cycle progression including CDKN1B.	0.902
17. NUP107 (Nuclear pore complex protein)	Required for the assembly of peripheral proteins into the NPC.	0.902
18. NDC1 (Nucleoporin)	Plays a key role in de novo assembly and insertion of NPC in the nuclear envelope. Required for NPC and nuclear envelope assembly, possibly by forming a link between the nuclear envelope membrane and soluble nucleoporins, thereby anchoring the NPC in the membrane.	0.902
19. NUP210 (Nuclear pore membrane glycoprotein)	Essential for nuclear pore assembly and fusion, as well as structural integrity.	0.901
20. NUP35 (Nucleoporin)	Can play the role of both NPC structural components and of docking or interaction partners for transiently associated nuclear transport factors.	0.901
21. POM121 (Nuclear envelope pore membrane protein)	Essential component of the nuclear pore complex (NPC). May be involved in anchoring components of the pore complex to the pore membrane.	0.900
22. POM121C (Nuclear envelope pore membrane protein)	Essential component of the nuclear pore complex (NPC). May be involved in anchoring components of the pore complex to the pore membrane.	0.900
23. NUP160 (Nucleoporins)	Involved in poly(A)+ RNA transport.	0.900
24. NUPL2 (Nucleoporin-like protein)	Required for the export of mRNAs containing poly(A) tails from the nucleus into the cytoplasm.	0.900
25. AAS (Nucleoporin)	Plays a role in the normal development of the peripheral and central nervous system.	0.900
26. GAK (Cyclin-G-associated kinase)	Involved in the uncoating of clathrin-coated vesicles by Hsc70 in non-neuronal cells.	0.874
27. STIP1 (Stress-induced-phosphoprotein)	Mediates the association of the molecular chaperones HSPA8/HSC70 and HSP90.	0857
28. EDEM3 (ER degradation-enhancing alpha-mannosidase-like protein 3)	Involved in endoplasmic reticulum-associated degradation (ERAD) of glycoproteins by proteasomes, by catalyzing mannose.	0.792
HspH2 Interaction Partners		
1. SNCA (Alpha-synuclein)	Induces fibrillization of microtubule-associated protein tau. Reduces neuronal responsiveness to various apoptotic stimuli, leading to a decreased caspase 3 activation.	0.965
2. C11 orf73	Acts as a specific nuclear import carrier for HSP70.	0.964
3. NUP62 (Nuclear pore glycoprotein)	Plays a role in mitotic cell cycle progression by regulating centrosome segregation, centriole maturation and spindle orientation. It might be involved in protein recruitment to the centrosome after nuclear breakdown.	0.944
4. RANBP2 (E3 SUMO-protein ligase)	Facilitates SUMO1 and SUMO2 conjugation, transport factor (Ran-GTP, karyopherin)-mediated protein import via the F-G repeat-containing domain which acts as a docking site for substrates. Component of the nuclear export pathway.	0.940
5. TPR (Nucleoprotein)	Essential for normal nucleocytoplasmic transport of proteins and mRNAs, plays a role in the establishment of nuclear-peripheral chromatin compartmentalization in interphase, and in the mitotic spindle checkpoint signaling during mitosis.	0.935
6. NUP37 (Nucleoporin)	Component of the Nup107-160 subcomplex of the nuclear pore complex (NPC) required for normal kinetochore microtubule attachment, mitotic progression and chromosome segregation.	0.929
7. OLR1 (Oxidized low-density lipoprotein receptor)	Mediates the recognition, internalization and degradation of oxidatively modified low-density lipoprotein (oxLDL) by vascular endothelial cells.	0.927
8. NUP155 (Nuclear pore complex protein)	Essential for embryogenesis. Nucleoporins may be involved both in binding and translocating proteins during nucleocytoplasmic transport.	0.925
9. NUP54 (Nucleoporin p54)	Component of the nuclear pore complex, a complex required for the trafficking across the nuclear membrane.	0.921
10. CCT2 (T-complex protein 1 subunit beta)	Molecular chaperone; assists the folding of proteins upon ATP hydrolysis. Known to play a role, in vitro, in the folding of actin and tubulin.	0.917
11. RAE1 (mRNA export factor)	Plays a role in mitotic bipolar spindle formation. Binds mRNA. May function in nucleocytoplasmic transport.	0.916
12. NUP107 (Nuclear pore complex protein)	Required for the assembly of peripheral proteins into the NPC. May anchor NUP62 to the NPC.	0.916
13. NUP214 (Nuclear pore complex protein)	May serve as a docking site in the receptor-mediated import of substrates across the nuclear pore complex.	0.915
14. NUP88 (Nucleoporins)	Essential component of nuclear pore complex.	0.915
15. NUP93 (Nuclear pore complex protein)	During renal development, regulates podocyte migration and proliferation through SMAD4 signaling.	0.914
16. AHSA1 (Activator of Hsp90 ATPase)	Activates the ATPase activity of HSP90AA1 leading to increase in its chaperone activity.	0.913
17. NUP153 (Nuclear pore complex protein)	Essential for normal nucleocytoplasmic transport of proteins and mRNAs. Involved in the quality control and retention of unspliced mRNAs in the nucleus.	0.912
18. NDC1 (Nucleoporin)	Plays a key role in de novo assembly and insertion of NPC in the nuclear envelope.	0.907
19. NUP205 (Nuclear pore complex protein)	Plays a role in the nuclear pore complex (NPC) assembly and/or maintenance.	0.907
20. NUP160 (Nuclear pore complex protein)	Involved in poly(A)+ RNA transport.	0.907
21. NUP50 (Nuclear pore complex protein)	Interacts with regulatory proteins of cell cycle progression.	0.906
22. NUP35 (Nucleoporin)	Can play the role of both NPC structural components and of docking or interaction partners for transiently associated nuclear transport factors.	0.903
23. AAAS (Nucleoporins)	Plays a role in the normal development of the peripheral and central nervous system.	0.903
24. FKBP4 (Peptidyl-prolyl cis-trans isomerase)	Immunophilin protein with PPIase. Plays a role in the intracellular trafficking of heterooligomeric forms of steroid hormone receptors between cytoplasm and nuclear compartments. Acts also as a regulator of microtubule dynamics by inhibiting MAPT/TAU ability to promote microtubule assembly.	0.903
25. NUPL2 (Nucleoporin-like protein)	Required for the export of mRNAs containing poly(A) tails from the nucleus into the cytoplasm.	0.903
26. NUP85 (Nuclear pore complex protein)	Required for spindle assembly during mitosis.	0.903
27. NUP188 (Nucleoporin)	May function as a component of the NPC.	0.903
28. NUP210 (Nuclear pore membrane glycoprotein)	Nucleoporin essential for nuclear pore assembly and fusion, nuclear pore spacing, as well as structural integrity.	0.902
29. POM121C (Nuclear envelope pore membrane protein)	Essential component of the nuclear pore complex (NPC).	0.902
30. POM121 (Nuclear envelope pore membrane protein)	Essential component of the nuclear pore complex (NPC).	0.902
31. GAK (Cyclin-G-associated kinase)	Involved in the uncoating of clathrin-coated vesicles by Hsc70 in non-neuronal cells.	0.893
32. CPS1 (Carbamoyl-phosphate synthase)	Involved in the urea cycle; plays an important role in removing excess ammonia from the cell.	0.891
33. CLPB (Caseinolytic peptidase B protein)	May function as a regulatory ATPase and be related to secretion/protein trafficking process.	0.889
34. AKT1 (RAC-alpha serine/threonine-protein kinase)	Regulates many processes including metabolism, proliferation, cell survival, growth and angiogenesis. AKT is responsible of the regulation of glucose uptake.	0.883
35. CCT5 (T-complex protein 1 subunit epsilon)	Known to play a role, in vitro, in the folding of actin and tubulin.	0.881
36. TP53 (Cellular tumor antigen p53)	Acts as a tumor suppressor in many tumor types; induces growth arrest or apoptosis depending on the physiological circumstances and cell type. Involved in cell cycle regulation as a trans-activator that acts to negatively regulate cell division by controlling a set of genes required for this process.	0.881
37. CCT3 (T-complex protein 1 subunit gamma)	Known to play a role, in vitro, in the folding of actin and tubulin.	0.856
38. EEF1A1 (Elongation factor 1-alpha 1)	Promotes the GTP-dependent binding of aminoacyl-tRNA to the A-site of ribosomes during protein biosynthesis. Forms a complex that acts as a T helper 1 (Th1) cell-specific transcription factor and binds the promoter of IFN-gamma to directly regulate its transcription and is thus involved importantly in Th1 cytokine production.	0.853
39. CCT4 (T-complex protein 1 subunit delta)	Known to play a role, in vitro, in the folding of actin and tubulin.	0.851
40. CCT6A (T-complex protein 1 subunit zeta)	Known to play a role, in vitro, in the folding of actin and tubulin.	0.848
41. NR3C1 (Glucocorticoid receptor)	Has transcriptional repression activity.	0.845
42. APAF1 (Apoptotic protease-activating factor 1)	Mediates the cytochrome c-dependent autocatalytic activation of procaspase 9 (Apaf-3), leading to the activation of caspase 3 and apoptosis.	0.841
43. CFTR (Cystic fibrosis transmembrane conductance regulator)	Regulation of epithelial ion and water transport and fluid homeostasis. Mediates the transport of chloride ions across the cell membrane.	0.836
44. SGT1	May play a role in ubiquitination and subsequent proteasomal degradation of target proteins.	0.820
45. OLA1 (Obg-like ATPase)	Hydrolyzes ATP, and can also hydrolyze GTP with lower efficiency.	0.820
Grp170 Interaction Partners		
1. PDIA4 (Protein disulphide-isomerase)	Belongs to the protein disulphide isomerase family.	0.993
2. SIL1 (Nucleotide exchange factor)	Functions as a nucleotide exchange factor for the ER lumenal chaperone HSPA5.	0.989
3. SEC63 (Translocation protein)	Required for integral membrane and secreted preprotein translocation across the endoplasmic reticulum membrane.	0.962
4. P4HB (Protein disulphide-isomerase)	May cause structural modifications of exofacial proteins. Inside the cell, seems to form/rearrange disulphide bonds of nascent proteins.	0.954
5. CALR (Calcium-binding chaperone)	Promotes folding, oligomeric assembly and quality control in the endoplasmic reticulum (ER) via the calreticulin/calnexin cycle. Interacts transiently with almost all of the monoglucosylated glycoproteins that are synthesized in the ER.	0.954
6. PDIA6 (Protein disulphide-isomerase A6)	Negatively regulates the unfolded protein response (UPR) through binding to UPR sensors such as ERN1, which in turn inactivates ERN1 signaling.	0.950
7. MANF (Mesencephalic astrocyte-derived neurotrophic factor)	Inhibits cell proliferation and endoplasmic reticulum (ER) stress-induced cell death (182 aa).	0.946
8. CANX (Calcium-binding protein)	May act in assisting protein assembly and/or in the retention within the ER of unassembled protein subunits. It seems to play a major role in the quality control apparatus of the ER by the retention of incorrectly folded proteins.	0.934
9. PDIA3 (Protein disulphide-isomerase A3)	Belongs to the protein disulphide isomerase family.	0.924
10. SEC61A1 (Protein transport protein)	Plays a crucial role in the insertion of secretory and membrane polypeptides into the ER. Required for assembly of membrane and secretory proteins.	0.906
11. CRELD2	Cysteine rich with EGF-like domains	0.901
12. SLC9C2	Sodium/hydrogen exchange; Involved in pH regulation.	0.876
13. ANKRD45	Ankyrin repeat domain-containing protein.	0.876
14. ERLEC1 (Endoplasmic reticulum lectin 1)	May function in endoplasmic reticulum quality control and endoplasmic reticulum-associated degradation (ERAD) of both non-glycosylated proteins and glycoproteins.	0.876
15. CLGN (Calmegin)	Functions during spermatogenesis as a chaperone for a range of client proteins that. are important for sperm adhesion onto the egg zona pellucida and for subsequent penetration of the zona pellucida.	0.868
16. DDOST (Dolichyl-diphosphooligosaccharide-protein glycosyltransferase)	Essential subunit of the N-oligosaccharyl transferase (OST) complex which catalyzes the transfer of a high mannose oligosaccharide from a lipid-linked oligosaccharide donor to an asparagine residue. Required for efficient N-glycosylation.	0.866
17. SEC61A2 (Protein transport protein)	Plays a crucial role in the insertion of secretory and membrane polypeptides into the ER. It is required for assembly of membrane and secretory proteins.	0.826
18. OS9	Functions in endoplasmic reticulum (ER) quality control and ER-associated degradation (ERAD).	0.803
19. SYVN2 (E3 ubiquitin-protein ligase synoviolin)	Component of the endoplasmic reticulum quality control (ERQC) system. Protects cells from ER stress-induced apoptosis.	0.786
20. SDF2L1	Stromal cell derived factor 2 like 1.	0.783
21. SEL1L	Plays a role in the endoplasmic reticulum quality control (ERQC) system. Plays a role in LPL maturation and secretion. Required for normal differentiation and survival of pancreatic cells.	0.783
22. CDNF (Cerebral dopamine neurotrophic factor)	Prevents the 6- hydroxydopamine (6-OHDA)-induced degeneration of dopaminergic neurons. Also prevents the degeneration of dopaminergic neurons.	0.777
23. EDEM3 (ER degradation-enhancing alpha-mannosidase-like protein 3)	Involved in endoplasmic reticulum-associated degradation (ERAD). Accelerates the glycoprotein ERAD by proteasomes.	0.773
24. SLC35B2	May indirectly participate in activation of the NF- kappa-B and MAPK pathways.	0.768
25. GAK (Cyclin-G-associated kinase)	Is involved in the uncoating of clathrin-coated vesicles by Hsc70 in non-neuronal cells.	0.752

Cut off score 0.75.

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
