# Peer review of "The Role of Non-Canonical Hsp70s (Hsp110/Grp170) in Cancer"

_cells, 2021, doi:10.3390/cells10020254_

Round 1

Reviewer 1 Report

The review provides a confused overview of the role of HSP110/Grp70 in cancer. The introduction is a continuation of general notions about cancer, which are useless. Table 1 is intended to give an overview of HSP110/grp70 family members with distinct functions and their related references but lacks to do so. Some parts seem to be the addition of references without logical flow. Concerning the part 3 (hap110 roles in cancer pathogenesis), reference 62 is not about HSPH1. The right reference is : Taguchi et al. PNAS 2019, HSP110 mitigates a-synuclein pathology in vivo. The authors never refer to the control of Wnt/b-catenin pathway by HSPH1, which is very surprising as the literature about HSPH1 in cancer is not large! In addition, the notion that HSP110 associates with IL-6 and TNFa is not very clear (line 201, page 8). The authors often give hypothesis that are not grounded enough like the one about a link with angiogenesis.

The most interesting part is about the unique proteomic signatures of HSP110 but is mostly based on preliminary data from the authors. Surprisingly, it is provided in a supplementary figure instead of a full table within the text.

Reviewer 2 Report

The authors have successfully targeted every important parameter of the topic pertaining to non-canonical Hsp70 involvement and interactions with oncogenic signaling pathways and have brilliantly discussed every subject of the matter using functional figures.

I do have some small indications/corrections that I would like to point out:

  1. NEF is used as an abbreviation without being mentioned or defined
  2. The manuscript has to be thoroughly re-read and corrected for some slight grammatical mistakes as some words are missing as well.
  3. Optional: The authors stressed throughout the whole manuscript on HSP110’s role as cellular buffers against proteostatic stress however dedicating a small paragraph that elaborates further on this topic would have been appreciated.
  4. Line 398 “to bind and block Hsp110 NBD and SBD segments possess potential in Hsp110-directed therapy.” You might want to add the word “both” or “simultaneously” to stress on the role of the small molecule inhibitor in blocking both the SBD and NBD at the same time. So, the sentence would become the following “to bind and block both Hsp110 NBD and SBD segments simultaneously possess potential in Hsp110-directed therapy.”

Author Response

Refer to attached document.

Reviewer 3 Report

The review article by Graham Chakafana and Addmore Shonhai is an interesting read worth of publication in Cells. I have several recommendations that aim to improve both clarity and content, as I detail on below:

  1. Referencing is often poor and needs to be fixed. In several instances it is either inaccurate or could use better citations to support their point. I exemplify below a few instances, but the authors should carefully check each citation.

Page 13: refs 122-124 do not support the statement: Two drugs targeting the SCD of Hsp70, targeted drug 2-phenylethynesulfonamide (PES) and the TKD-motif directed peptide inhibitor, cmHsp70.1 have entered clinical trials stage [122, 123, 124]. Also, these are inhibitors or drug candidates (if indeed in clinic now) but not yet drugs (i.e. not approved nor used for human disease). Please rectify.

Page 4. This sentence is important “Thus, Hsp110/Grp170 are more effective holdase chaperones than their canonical Hsp70 counterparts [37]. Hence canonical Hsp70s serve as a refoldase while Hsp110/Grp170 members are buffers against proteostatic stress [37].” yet I fail to understand how ref 37 is best to support it.

Page 3. It is unclear to me how ref 26 supports this statement: “Remarkably, Hsp70 and Hsp27 have both been shown to interact directly with protein intermediates of the apoptosis pathway [25-26].”

[26] Lanneau, G.S.; Argenta, P.A.; Lanneau, M.S.; Riffenburgh, R.H.; Gold, M.A.; McMeekin, D.S.; Webster, N.;Judson, P.L. Vulvar cancer in young women: demographic features and outcome evaluation. American journal of obstetrics and gynecology, 2009 200(6), pp.645-e1.

Page 2. Ref 21 supports changes in complexation not expression as a contributing factor in the role of these proteins in tumor progression. “Therefore, elevated Hsp levels are associated with tumour progression [21].”

  1. The article gives a one-sided view on chaperones in cancer whereby overexpression is the sole culprit in their involvement in cancer processes. This view ignores recent advances in the field linking complexation changes, that are independent of expression, to their transforming activity. The authors should read ref 21 above, as well as later articles on the subject, especially PMID: 29795326; PMID: 31668946; PMID: 32610141, to address this point in the revised manuscript

  1. Minor: edits are needed to fix a few typos. Examples:

Page 3 and several other instances. Space and punctuation mark missing between the ref bracket and text “cancer [17-20]Due”

Page 4: “In comparison to Hsp110, Grp170 is exhibits a unique”

Page 8. reported that Hsp70s, associates with

Page 9: involvement of Hsp110/rp170 in cancer

roles in chaperoning proteins involves in the different cancer

  1. Please check citation/ref style
  2. In my opinion it would be useful to include a few citations sending the reader to recent review articles that may provide complementary insights on the topic

Author Response

Refer to attached document.

Round 2

Reviewer 1 Report

I am pleased that the authors have taken a strong attention to my comments and other reviewers as well. The manuscript has been strongly improved. 

I still have the feeling that the general introduction about HSP in cancer is not necessary altough it has been strongly reduced in the new version.